# Human Germ Cell Tumors are Developmental Cancers: Impact of Epigenetics on Pathobiology and Clinic

**DOI:** 10.3390/ijms20020258

**Published:** 2019-01-10

**Authors:** João Lobo, Ad J. M. Gillis, Carmen Jerónimo, Rui Henrique, Leendert H. J. Looijenga

**Affiliations:** 1Cancer Biology and Epigenetics Group, Research Center of Portuguese Oncology Institute of Porto (GEBC CI-IPOP), R. Dr. António Bernardino de Almeida, 4200-072 Porto, Portugal; joaomachadolobo@gmail.com (J.L.); carmenjeronimo@ipoporto.min-saude.pt (C.J.); henrique@ipoporto.min-saude.pt (R.H.); 2Department of Pathology, Portuguese Oncology Institute of Porto (IPOP), R. Dr. António Bernardino de Almeida, 4200-072 Porto, Portugal; 3Department of Pathology and Molecular Immunology, Institute of Biomedical Sciences Abel Salazar, University of Porto (ICBAS-UP), Rua Jorge Viterbo Ferreira 228, 4050-513 Porto, Portugal;; 4Laboratory of Experimental Patho-Oncology (LEPO), Josephine Nefkens Building, Erasmus MC, Department of Pathology, University Medical Center, Cancer Institute, Be-432A, PO Box 2040, 3000 CA Rotterdam, The Netherlands; a.gillis@erasmusmc.nl; 5Princess Máxima Center for Pediatric Oncology, Heidelberglaan 25, 3584 CS Utrecht, The Netherlands

**Keywords:** biomarkers, development, epigenetics, germ cell cancer, methylation, microRNAs, testicular cancer

## Abstract

Current (high throughput omics-based) data support the model that human (malignant) germ cell tumors are not initiated by somatic mutations, but, instead through a defined locked epigenetic status, representative of their cell of origin. This elegantly explains the role of both genetic susceptibility as well as environmental factors in the pathogenesis, referred to as ‘genvironment’. Moreover, it could also explain various epidemiological findings, including the rising incidence of this type of cancer in Western societies. In addition, it allows for identification of clinically relevant and informative biomarkers both for diagnosis and follow-up of individual patients. The current status of these findings will be discussed, including the use of high throughput DNA methylation profiling for determination of differentially methylated regions (DMRs) as well as chromosomal copy number variation (CNV). Finally, the potential value of methylation-specific tumor DNA fragments (i.e., *XIST* promotor) as well as embryonic microRNAs as molecular biomarkers for cancer detection in liquid biopsies will be presented.

## 1. Introduction: Germ Cell Tumors in General

### 1.1. Epidemiology

A number of different cell types can be found in the testis (germ cells, Sertoli cells, Leydig cells, mesenchymal cells, mesothelial cells, among others); thus, and despite being a relatively small organ, the testis may give rise to a large variety of neoplasms. Nonetheless, more than 95% of testicular neoplasms are derived from germ cells arrested in their differentiation—the testicular germ cell tumors (TGCTs) [1]—meaning that global epidemiological trends for testicular cancer mostly refer to this group of neoplasms. Still, germ cell tumors (GCTs) are not simply a single class of neoplasms; instead, they represent a heterogeneous array of tumor entities, arising either in the gonads (both male and female) or being extragonadal (developing from germ cells arrested in their migration along the midline of the body), reflecting a complex and development-related tumor model, each subclass showing its peculiarities and specific epidemiology. The *European Cancer Registry-Based Study on Survival and Care of Cancer Patients (EUROCARE)* reports markedly distinct age-adjusted incidence rates of GCTs in Europe for males and females (64 per 1,000,000 versus 4 per 1,000,000, respectively) [2]; besides these global differences in incidence, there are histological divergences among testicular and ovarian neoplasms: in males most tumors are seminomas (SEs) and most non-seminomatous tumors (NSTs) are mixed forms, while in women most tumors are NSTs and mixed forms are the exception. TGCTs show a ‘bell-shaped’ distribution of cases with a peak around 30 years, with SEs overall occurring 10 years later than NSTs. In the United States incidence rates were of 56 per 1,000,000 and of only 10 per 1,000,000 in Caucasian and African-American males, respectively [3]. Only 4% of GCTs are extragonadal, located at central nervous system, mediastinum, retroperitoneum, and pelvis, and the majority correspond to NSTs. Incidence of GCTs overall has been increasing, especially at the expense of TGCTs. Five-year survival is better for gonadal GCTs when compared to extragonadal GCTs [2].

Overall, testicular cancer is not a common disease, ranking as only the 21st most incident neoplasm in men worldwide, in 2018 (with 71,105 new cases and an age standardized rate of 1.7 per 100,000). It is neither a frequent cause of cancer-related deaths, being the 27th most lethal malignancy in men (with 9,507 estimated deaths in 2018). However, a closer look at the figures shows it also represents the most incident and most prevalent neoplasm in males aged 15–39 years old, at global level, with an age-standardized incidence rate of 2.7 per 100,000 in 2018 and a five-year prevalence of 150,377 cases. Moreover, incidence is on the rise in most populations [4], with a total of 85,635 new cases expected for 2040 (14,530 more that in 2018, representing a 20.4% increase), and a total of 13,288 estimated deaths (3781 more than in 2018, a 39.8% increase), according to *Globocan 2018* predictions [5]. Similar data is replicated by the *Surveillance, Epidemiology, and End Results Program (SEER)* database; despite representing only 0.5% of all new cancer cases in the United States, incidence rates have been rising 0.8% per year over the last 10 years and the number of new cases of testicular cancer was 5.7 per 100,000 males per year according to 2011–2015 registries. Median age at diagnosis is 33 years-old (25 years for NSTs, 35 years for SEs) and an intermediate 30 years for NSTs with a SE component). The majority (68%) of patients are diagnosed with localized disease (>80% of SEs and >60% of NSTs present with clinical stage I disease). Metastases emerge in 15% and 20% of stage I SE and NST patients, respectively, within a period of two to three years [1]. Still, five-year survival is outstanding (95.3%), even for patients diagnosed with distant metastases overall (73.7%) [6,7,8]. Importantly, variation in testicular cancer incidence rates worldwide is remarkable (29-fold variation, being higher in Europe, Australia and the United States), and mortality-to-incidence ratio is reported to be higher in underdeveloped regions of the globe, probably due to less access to proper healthcare facilities, diagnostic tools, and multimodal treatments [9]. This information is summarized in Table 1 below.

All in all, there are a number of reasons to remain focused on TGCTs: besides the rising incidence in part explained by Western lifestyle, about 15–20% of patients with disseminated disease experience disease recurrence (with late relapses displaying poor prognosis); in spite of exquisite sensitivity to cytotoxic agents, resistance to cisplatin treatment eventually emerges in some cases, by still elusive mechanisms; and also the diagnosis of cancer in such young patients (with long life-expectancy) who undergo chemo and radiotherapy raises concerns over quality of life, fertility and enduring treatment-related side effects, such as the emergence of second tumors and cardiovascular disease, and merit proper action to prevent them [10,11,12,13,14,15].

### 1.2. The ‘Genvironmental’ Model

TGCTs constitute a formidable example of how genetic and environmental risk factors can synergistically potentiate malignant transformation, in a so-called ‘genvironmental’ model [16].

#### 1.2.1. Genetic Risk Factors

Genetic factors play an important role in TGCTs genesis; in fact, they contribute to more than 40% of TGCTs (the third highest rate among all cancer types) [17,18,19]. Familial risk is one of the highest in cancer: having a brother or father diagnosed with a TGCT increases the risk eight-to-ten and four-to-six times, respectively; the risk in monozygotic and dizygotic twins is increased 76 and 35 times, respectively [20,21,22]. Also, there is an increased risk for developing cancer in the contralateral testis, which further strengthens the influence of genetic factors and may justify performing biopsies of the contralateral testis for identifying germ cell neoplasia in situ (GCNIS) [23,24]. Familial TGCT (FTGCT) is indeed nowadays well recognized [25], as is the risk association with specific variants of disorders of sex development (DSDs), also referred to as difference of sex development [26,27,28,29,30]. Still, more than 90% of TGCT patients end up showing no family history of TGCT. Evidence points towards the influence of various autosomal recessive low-penetrance susceptibility genes and polymorphic gene variants. Importantly, a model of true interplay between environmental and genetic factors is the most likely scenario in TGCTs the ‘genvironment’. This model fits TGCTs genesis and places epigenetic deregulation as the perfect culprit mechanism for mediating this environment-genetics interaction and for explaining clinical findings that compose the TDS and that associate with increased risk of TGCTs [16,31,32]. Over the years, a great effort (in the form of genome-wide association studies (GWAS) and various metanalyses) has been put towards characterizing susceptibility alleles for TGCTs emergence and, to date, a number of these have been determined, the ones showing the highest odds ratio thus far being the *KITLG*-related [33,34,35,36]. Recently, a couple of metanalyses have extended the number of susceptibility risk loci to 44. Importantly, these studies have put in evidence the polygenic character of the TGCT tumor model, identifying candidate culprit single nucleotide polymorphisms (SNPs) implicated in several distinct pathways, from the well-known *KIT-KITL* to DNA damage repair (namely *RAD51C* and *BRCA1*), from sex determination/germ cell specification (namely *DMRT1*, *ZFPM1*, and *PRDM14*) to apoptosis/cell cycle (including *GSPT1* and *CHEK2*), from telomere maintenance (including *TERT* and *ATFIP*) to centrosome organization/microtubule assembly (*TEX14*, *PMF1*, and *CENPE*) [17,19,37]. Additionally, novel susceptibility markers have been pointed out by non-GWAS studies, such as *LRRC50*, being a member of cilia-microtubule genes, also known as *DNAAF1* [38,39].

#### 1.2.2. Environmental Risk Factors

Environmental risk factors can be categorized into distinct groups: the internal, less-modifiable, developmental/medical risk factors and the external, modifiable, post-natal risk factors. 

##### Internal Risk Factors

Overall, one can say that features comprising testicular dysgenesis syndrome (TDS) increase the risk of TGCTs [40]; indeed, the most consistent risk factor for TGCT development is cryptorchidism, increasing the risk in about five-fold [41], and other disorders such as hypospadias, testicular atrophy, inguinal hernia, and impaired spermatogenesis are risk factors as well [42]. Other risk factors include perinatal factors, such as low and high birthweight, decreased gestational age, maternal bleeding, and low parity. The unifying state among some of these factors is fetal exposure to estrogens and anti-androgens, resulting in disruption of endogenous hormone signaling prenatally by exposure to xenobiotics, leading to undervirilization of the male embryo in utero [43].

##### External Risk Factors

Other less established and not so well understood risk factors act postnatally; the contribution of these factors has been assessed in several studies, achieving different and sometimes opposing conclusions. However, although TGCTs emerge early in life, a potential role of these postnatal risk factors cannot be ignored. These include diet (high in fat and dairy products, putting natural or synthetic hormones as culprits), low physical exercise, environmental/occupational exposures (with increased risk in firefighters and metal, leather, and agricultural workers) and also testicular trauma [44]. Most studies found no association with body mass index, while most describe an association with tall stature. Late onset of puberty has been proposed to decrease risk. Marijuana smoking is of particular interest, since it preferentially associates with NST histology, suggesting a role in the reprogramming process [45,46]. The reason of the other risk factors to be predominantly acting during intra-uterine development compared to marijuana use remains to be elucidated.

#### 1.2.3. Interplay between Environmental and Genetic Risk Factors: The ‘Genvironment’

An interplay between the aforementioned genetic and environmental factors has been demonstrated to occur in various studies on TGCTs, with epigenetics serving as a bridge between these two aspects of the disease. This is illustrated, for instance, by the spectrum of clinical manifestations that is TDS. There is an association between infertility and TGCT risk, and also GCNIS lesions are found more often in infertile men when compared to the remaining population, reflecting this link between (in)fertility, andrological/urological abnormalities and TGCT risk, the latter being modified by a complex array of ‘genvironmental’ modifiers.

A summary of the role of these risk factors is depicted in Table 2.

### 1.3. Classification

The first attempt to classify the various subtypes of TGCTs was accomplished by Friedman and Moore in 1946, when they organized tumors into four groups: SEs, embryonal carcinomas (ECs), teratomas (TEs), and teratocarcinomas [47]. Since then a multitude of classifications have been proposed, reflecting a progressive better understanding of TGCTs tumorigenesis. The most recent 2016 *World Health Organization (WHO)* Classification is settled on the concept of GCNIS, and recognizes two major types of tumors: the GCNIS-related postpubertal-types, corresponding to SEs and NSTs, and the GCNIS-unrelated neoplasms, comprising both prepubertal-type tumors (i.e., TEs and yolk sac tumors (YSTs) and spermatocytic tumor (ST), previously known as spermatocytic seminoma) [1,48]. With this classification we have witnessed a transition from an essentially morphological system into one that reflects TGCTs pathogenesis and the developmental potential of the cells from which they derive. It is, indeed, an improvement from the former 2004 classification, which considered morphologically similar but very distinct entities under the same group, such as the former historically erroneous designation “spermatocytic seminoma” [49].

Novel and more profound understanding of mechanisms regulating embryogenesis and implicated in germline allow for postulating a developmental model for GCTs in general, based on a biologically plausible, sound and clinically relevant basis for classification. This way, Oosterhuis and Looijenga proposed a new broad classification of GCTs, according to their developmental potential (Table 3) [50]. The classification considers seven types of GCTs, adding type 0 (included and parasitic twins) and type VI (arising from somatic cells which are induced to pluripotency) to the previous five groups already acknowledged by these authors [51].

In this review, we will be focusing especially on type II GCTs occurring in the testis, which are the most frequent, the most histologically and clinically diverse and the most extensively studied [18]. However, some of the topics to be discussed are also of relevance for some of the other types of GCTs.

## 2. Pathobiology of Germ Cell Tumors and their Developmental Potential

### 2.1. Normal Physiology of Embryonic and Germ Cell Development

In a simple way, for full human development to take place, the omnipotent zygote needs to undergo a program of successive restrictions of pluripotency, which is tightly regulated. However, and for assuring maintenance of the species, totipotency of the germ cell lineage must be essentially preserved [52].

Primordial germ cells (PGCs), the precursors of the various maturation stages of the germ cell lineage, undergo a process of specification, proliferation and, finally, migration along the midline of the body (explaining the topography of extragonadal GCTs), from the yolk sac, via the hindgut and towards the genital ridge, where sex determination subsequently occurs. Once arriving at the genital ridges they are referred to as gonocytes, which further differentiate into pre-spermatogonia or oocytes depending on the chromosomal constitution, on the action of the major player *SRY* gene and on the gonadal microenvironment, through a complex downstream signaling cascade [53,54]. During these processes of migration and maturation, an epigenetic ‘reset’ is warranted, in the form of early global DNA demethylation. This marks the start of the genomic imprinting (GI) cycle. GI (uncovered by *Solter* and *Surani* in 1984 [55,56,57,58]) refers to somatic inheritance of epigenetic marks, independently from the Mendelian laws of inheritance; in other words, it allows for some genes to be expressed depending on the parent (maternal or paternal) of origin, thanks to parental-specific DNA methylation and histone modifications [59]. A biparental GI is necessary for full development of the zygote, as is a specifically methylated intact genome with X chromosome inactivation in females. After fertilization takes place, the zygote receives a haploid set of paternally imprinted chromosomes from the father and a haploid set of maternally imprinted chromosomes from the mother, originating a diploid zygote with biparental GI pattern, which is replicated throughout full development of the embryo. The zygote undergoes two waves of demethylation (active demethylation in the paternal pronucleus and passive demethylation of the maternal genome), followed by remethylation with onset in the inner cell mass of the developing embryo. In the germline, however, GI needs to be early erased during PGCs migration, by means of demethylation of CpG sites, allowing for genesis of germ cells with no parental-specific epigenetic modifications. Finally, later on, when the maturing germ cells arrive at the genital ridges and reach mitotic (in males, spermatogenesis) or meiotic (in females, oogenesis) arrest, a restoration of the uniparental sex-specific (maternal or paternal) GI occurs, by *de novo* methylation of relevant target gene, closing the cycle [60,61,62,63,64] (Figure 1). Besides GI, X chromosome reactivation can be also considered a marker of developmental stage which occurs in female germ cells before oogenesis starts [65,66,67].

A full understanding of all stages of embryonic development and germ line establishment is the cornerstone of conceiving a unifying model for the pathogenesis of GCTs, where the developmental potential of each tumor entity is determined by the developmental state of the originating cell. This is an appealing model, as GCTs are truly in the crossroads between cancer and developmental biology [50,68]. For instance, it is easy to conceive that type III GCTs (which refer to STs) have a developmental potential restricted to postpubertal, premeiotic, spermatogenic cells (spermatogonium/spermatocytes) and display a GI pattern that is partially to completely paternal, while on the other hand type IV GCTs (which include dermoid cysts) derive from activated oocytes/ovum incapable of supporting the development of extraembryonic tissues, presenting with a GI status that is partially to completely maternal. Also, type V GCTs (which refer to hydatidiform moles), derive from an empty ovum that is subsequently fertilized by one sperm, hence presenting an exclusively paternal GI pattern, having no ability to give rise to somatic tissues.

### 2.2. Type II Germ Cell Tumors of the Testis

#### 2.2.1. Developmental Potential

According to the proposed and above-mentioned model [50], errors in regulation of the developmental potential of embryonic stem and early PGCs may give rise essentially to extragonadal tumors early in life, whereas deregulation of the developmental potential already in the germline originates a multitude of tumors preferentially located in the gonads which are primarily diagnosed after childhood. In practice, this unifying model for GCTs enables them to be classified according to their developmental capabilities, and categorized in groups sharing common features, such as epidemiology, anatomical site distribution, cytogenetic abnormalities and epigenetic (de)regulation mechanisms like GI profile.

Regarding type II GCTs, the same developmental proximity is maintained; they have the broadest developmental potential (being in fact omnipotent) and comprise two major groups of neoplasms, the SEs (also referred to as dysgerminoma and germinoma in the ovary and in the brain, respectively), which derive from PGCs/gonocytes delayed in their maturation and constitute the so-called ‘default pathway’; and the NSTs, which arise when a neoplastic PGC/gonocyte (either from GCNIS or overt SE) undergoes a process of reprogramming, leading to the formation of a totipotent EC cell. In turn, the latter has the capacity of originating tumor components representative of all lineages, including the extra-embryonal YST and choriocarcinoma (CH), and the somatic-derived TE, which includes somatic tissues from the three germ layers in varying degrees of maturation (i.e., pluripotent). A combination of any of these components leads to the formation of mixed tumors [69,70,71] (Figure 2).

In the early developmental stage of type II TGCTs, there is a continuum from a delay in maturation of gonocytes, pre-GCNIS and GCNIS; in the latter stage the neoplastic gonocytes are located in a territory known as spermatogonial niche, and consistently fail to switch off and hence express OCT3/4 (a transcription factor expressed both in normal PGCs and embryonic stem cells as well as in their neoplastic counterparts, SEs and ECs), usually in conjunction with expression of TSPY and presence KITLG (illustrating once again that TGCTs are developmental cancers, not due to accumulation of mutations but instead thanks to deregulation of expression of critical differentiation-related proteins) [72,73]. GCNIS (first discovered in 1972 by Skakkebaek [74]) virtually always progresses to overt TGCT (50% at five years and 70% at seven years [75]), passing through an intermediate stage in the ‘default pathway’ of intratubular SE before turning into a fully invasive SE. An escape from the ‘default pathway’ occurs through reprogramming of a SE cell, either intratubular or invasive, leading to emergence of an EC cell and originating NST components [76]. It remains to be proven whether all GCNIS will become invasive tumors.

#### 2.2.2. Brief Pathogenetic Overview

Type II GCTs are always malignant. Most (>90%) arise in the testis (GCNIS-related postpubertal-type tumors), the rest occurring in dysgenetic gonads and in extragonadal sites [3]. Type II TGCTs are the most common malignancies in Caucasian males aged between 25 and 45 years-old in Western populations; of these, slightly more than 50% are pure SEs, the second most common being mixed tumors [4]. About 25% of these tumors are expected to be due to familial susceptibility [77]. Also, they are consistently peritriploid, being characterized by gains of the short arm of chromosome 12 (frequently in the form of isochromosome 12p, [i(12p)]) [78,79], which makes sense considering that these tumors undergo a prolonged period of karyotype evolution since the intratubular phases of tumor development. Polyploidization, in addition to a hypomethylated genome, contribute to chromosomal instability in these neoplasms, which further drives tumor progression [80]; however, mutations and amplifications of oncogenes are rather rare in TGCTs, with *KIT* mutation being the most common, especially in SEs and in bilateral cases [81,82,83]. In fact, recent work by Dorssers et al. [84] has showed, by use of whole genome and targeted-sequencing, that NSTs are initiated by genome duplication, followed by chromosome copy number alterations in cancer stem cells, with very low accumulation of somatic mutations, even in cases resistant to therapy. Metastatic tumors show, in fact, very little overlap with the originating primary tumor and precursor lesions, meaning that treatment of recurrences deserve therapies targeted at their specific molecular landscape.

While differences in incidence across several regions of the globe demonstrate the relevance of environmental factors in their genesis, contrasting incidence rates in distinct ethnic groups of the same populations underline the contribution of genetic factors—in sum, the role of the aforementioned ‘genvironmental’ model [85].

## 3. Taking Advantage of the Developmental Model: Biomarkers for Clinical Implementation

### 3.1. Use of High-Throughput Methodologies

All seven subtypes of GCTs are derived from germ cells in distinct maturation stages and with distinct methylation profiles. A way to both support and study this tumorigenesis model has necessarily to pass through implementation of high-throughput, genome-wide methodologies. The implementation of techniques such as whole genome sequencing (WGS), whole exome sequencing (WES), targeted sequencing, RNA-sequencing (RNA-seq), miR profiling, methylation profiling/arrays and proteomics, and their combination (with proper exclusion of contaminants such as blood cells, stromal cells, immune cells, and compensation for other confounders), will open the door for clarifying findings of previous studies and for uncovering novel disease biomarkers. Some of these methodologies have already been employed in some studies and will be discussed in the following section.

### 3.2. The Role of Epigenetics

Epigenetics encompasses an array of processes that change gene expression without altering the DNA sequence, leading to a change in phenotype without changing the genotype. It comprises covalent modifications of DNA (such as DNA methylation), histone variants, histone post-translational modifications, and non-coding RNAs (ncRNAs). DNA methylation, one of the most studied mechanisms, occurs by addition of methyl groups to the fifth carbon of cytosines, occurring preferentially at CpG sites, which are unevenly distributed in the genome—being concentrated in the so-called CpG islands. Differential methylation of gene promoters ultimately affects gene expression. Similarly, a number of ncRNAs are involved in the dynamic and environmentally sensitive regulation of gene expression. These molecules are known to interact (directly or indirectly) with the other established epigenetic mechanisms and can also directly interfere with messenger RNA (mRNA); this way, they can be seen as an extension of the complex epigenetic network, establishing important bridges between related modifications and truly influencing gene expression [86,87,88]. In this review, we will be focusing on methylation and a subtype of ncRNAs—the microRNAs (miRs).

#### 3.2.1. Methylation-Based Biomarkers Relating to the Developmental Model

X-chromosome inactivation (also known as lyonization [89]), a process limited to germ cells in the male, opens the way for uncovering novel biomarkers in TGCTs, namely concerning X-inactive specific transcript (*XIST*) gene, which is mapped to the X-chromosome inactivation center (XIC). In females, *XIST* (a long non-coding RNA [lnc-RNA] encoded from one of the X chromosomes, which is triggered when in *cis*) is responsible for inactivation (by inducing methylation of X-linked genes) of the extra X chromosome (compensating for the increased dosage of X-linked genes when compared to male individuals) [90,91]. The fact that TGCTs of the adult testis (SEs and NSTs) often show supernumerary X chromosomes suggests that *XIST* might also be expressed in these tumors. Indeed, both GCNIS and overt TGCTs have been shown to express *XIST* (contrarily to lower expression in normal testicular parenchyma with active spermatogenesis), which effectively rendered methylation of the X-chromosome gene coding for androgen receptor (AR) [92]. One might say, then, that X inactivation by *XIST* allows the tolerance of an excessive gain of X chromosomes in TGCTs, particularly in SEs, which show higher expression than NSTs, meaning that this differentiation-dependent mechanism of inactivation is indeed preserved and functional in these neoplasms [93]. In this regard, Looijenga et al. findings [92] reflect, once again, a developmental model for TGCTs, with GCNIS and SEs showing *XIST* expression still not accompanied by methylation of the inactive X chromosome (a status typical of early germ cells), with more differentiated NSTs exhibiting both *XIST* expression and methylation of the inactive X chromosome (similar to female and extraembryonic tissues), and finally ECs displaying an intermediate pattern between the two.

TGCTs are stated to be hypomethylated when compared to other somatic-derived cancers, which makes it difficult to uncover methylation-based biomarkers for TGCTs, especially for more undifferentiated forms such as SEs. However, demethylated-based biomarkers might constitute a good strategy for these tumors [94,95,96]. In TGCTs, *XIST*, which has de advantage of being expressed specifically in male germ cells, is frequently hypomethylated at its 5′ end independently of its expression (but more in a differentiation-dependent manner), while in somatic cells its expression is regulated by methylation of its promoter (hypermethylation blocks *XIST* expression, resulting in an active X chromosome). In this line, Kawakami et al. [97] very elegantly characterized the 5′ end of *XIST* by bisulfite sequencing and identified regions I to VI, with 56 CpG sites. In their work, region IV was found to be the most promising (consistently demethylated), and primers for the respective methylated and unmethylated sequence were designed for conventional polymerase chain reaction (PCR). Region IV was frequently unmethylated in TGCTs, especially in SEs and patients with advanced disease, and no demethylated signals were detected in somatic cancers in males (serum samples of kidney and bladder cancer patients). The authors explained the finding of some methylated signals in TGCTs as probably due to contamination with other cells. Investing in exploring these demethylated *XIST* fragments as biomarkers for TGCTs diagnosis in liquid biopsies is promising, especially if performed in larger cohorts and if novel detection methods with higher sensitivity and specificity are employed. These unmethylated fragments might prove very useful in the future for follow-up of SE patients, for which no available and reliable marker exist thus far [98]. In another study Ushida et al. [99] analyzed the methylation status of DNA repetitive elements in TGCTs, including *LINE1* and Alu repeats located at the 5′end of both E-Cadherin (*CDH1*) and *XIST*. By use of bisulfite sequencing and combined bisulfite restriction analysis (COBRA), the authors compared the (de)methylation profile of both TGCT cell lines, TGCT human tissues and somatic-derived cancers (kidney cancer cell lines and testicular lymphoma tissues), showing that *LINE1* was highly hypomethylated in both SEs and NSTs; on the other hand, the two chosen Alu elements were differentially methylated between SE and NST samples (being mainly hypomethylated in the former and methylated in the latter). Despite somatic-derived neoplasms also exhibited partial demethylation of these regions (perhaps reflecting the commonly accepted hypomethylation pattern of DNA repetitive elements in cancers in general [100,101]), the degree of demethylation was not as pronounced as in TGCT samples, hypothesizing that the genesis of these tumors and the mechanisms underlying maintenance of demethylation of repetitive elements might differ among somatic and germ cell-derived neoplasms. Indeed, since PGCs undergo GI erasure (including erasure of methylation in *LINE1* and Alu repeats [61]) and since SEs are composed of cells resembling PGCs/gonocytes, the (de)methylation profile observed in SEs might be specifically related to this resemblance and not to global demethylation observed in other cancer subtypes. Finally, the authors reported partial demethylation also in non-cancerous testicular parenchyma adjacent to TGCTs (without evidence of GCNIS); this somewhat unexpected finding is possibly due to epigenetic abnormalities related to defects in spermatogenesis and those occurring in stromal cells, similar to previous findings in gastric cancer [102]. These findings might point towards an impact of field-cancerization, with cancer-associated stromal cells sharing epigenetic changes with the accompanying tumor epithelial cells, and being biologically different from normal stroma unrelated to the tumor bulk. Similarly, in testes with impaired spermatogenesis, epigenetic aberrations (including demethylation) might have already taken place, despite no evidence of a germ cell lesion (invasive or precursor).

The promoter methylation status of several candidate genes has been assessed in various individual studies on TGCTs (reviewed in [103]), both in patients’ tissue and plasma cohorts and cell lines [104,105,106,107,108,109,110,111,112,113,114,115,116,117,118,119]. Some focused on genes involved in stages of embryonic development (such as *CRIPTO* [105] and *OCT3/4* [116]), others on genes having a tumor-suppressor role in a variety of neoplasms (such as *RASFF1A* [107,120]), others coding for DNA repair proteins (such as *MGMT* [109]) and others explored cancer-testis antigens (CTAs, such as *PRAME* [119]). Studies differ largely in methodologies employed (and related sensitivity and specificity), patient selection and samples studied (including distinct control samples and proportion of tumor subtypes), so they should be compared with caution. Some of these genes were indeed included in the DMRs picked up by genome-wide analyses [121], shedding more light on the findings of these studies.

Following the evidence that germ cells show transient erasure of GI, Killian et al. [122] explored the GI status of TGCTs by use of genome-wide DNA methylation analysis. In their study, the authors used only pure type II TGCT forms and neighboring testicular tissue without evidence of GCNIS. Besides performing somatic copy number aberrations (SCNA) analysis (requiring the gain of the short arm of chromosome 12 as an inclusion criteria), the authors introduced an adjunct technique to methylation profiling (450K Infinium bead-array), by using lymphoid compensation (LC). This lymphoid-compensated global methylation assessment might indeed prove very useful, as SEs, and sometimes also NSTs, are characterized by a prominent lymphoid infiltrate which contaminates the sample and confounds the results, possibly explaining the non-complete erasure observed in this tumor subtype (in which almost complete erasure would be expected). The authors confirmed, indeed, that LC resulted in shifting the peak methylation of SEs selectively close to zero, unmasking the true and whole erasure of these tumors. This way, the authors uncovered a hypomethylation locus which proved to be consistently present in TGCTs (both SEs and NSTs), corresponding to hypomethylation of the *DPPA3* (also known as *STELLA*) gene promoter. This maternal-effect gene implicated in protecting parental imprints from erasure in the post-fertilization demethylation process, is expressed in both TGCTs and PGCs [123], and was shown to be hypomethylated in TGCTs (irrespective of the histology—SEs or NSTs—reflecting the pattern observed in PGCs), but always (hyper)methylated in all tested somatic tissues. Interestingly, this promoter shows no CpG islands. This pattern of erasure and expression is maintained despite the occurrence of de novo methylation, meaning that *DPPA3* escapes differentiation-related methylation. This points out that *DPPA3* might constitute a promising biomarker for TGCTs. Also in that study, methylation patterns of neighboring tissue were dependent on spermatogenic proficiency. Analysis of GI revealed hypermethylation of *HM13* in NSTs and subtype-specific hypermethylation of *H19* in TEs, while SEs (like GCNIS) were globally GI-erased, implying that focal methylation observed in NST samples might occur de novo after erasure, as suggested in other studies [124]. This might be of interest in the prediction of development of residual mature TE in case of patients with NST treated with chemotherapy [125]. Additionally, the authors explored differentially methylated genes among the various pure tumor entities: SEs, ECs, YSTs, CHs, and TEs.

In another study, Rijlaarsdam et al. [121] extensively profiled 91 GCTs (of subtypes I–IV, including males and females) and four GCT cell lines (representative of type II GCTs) using the HumanMethylation450 BeadChip (450K array, Illumina). After data processing (assuring exclusion of confounders such as cross hybridization, SNPs, poor probe performance) a total of 437,882 valid probes were employed, with additional annotation including small nuclear RNAs (snRNAs) and miRs, repetitive elements and imprinted segments. Ultimately, differentially methylated genomic regions were uncovered, allowing for clustering tumor subtypes according to their methylation profile. For instance, the most remarkable methylation differences were found between the cluster composed of SEs, dysgerminomas and STs (found to be globally hypomethylated) and the one composed of ECs, NSTs, and type I TEs (hypermethylated). Methylation profiling also allowed for discrimination of individual tumor subtypes. Interestingly, the analysis depicted little similarity between GCT human tissues and GCT-representative cell lines, which warrants caution when employing these in vitro models in methylation-based analyses. Furthermore, a number of GCT methylation-related genes were confirmed and/or uncovered with these analyses. With this study, by studying DMRs in a genome-wide manner, the authors truly provided insight into GCTs biology and developmental genesis, further supporting a developmental model for these tumors. In yet another study, Noor et al. [126] explored methylation and expression profiles of GCT cell lines using the same HumanMethylation450 BeadChip (450K array, Illumina) and the Affymetrix GeneChip Human Genome U133 Plus 2.0 Array, disclosing that the hypermethylation status observed in YSTs is localized to certain CpG islands in a small proportion of genes (CpG island methylator phenotype), whereas for ECs and TEs methylation was more disperse. With this study the authors identified a wide list of genes differentially methylated between different cell lines and an inverse correlation with respective expression was established.

Cheung et al. [111] further studied epigenetic changes of ECs. In their study, they have profiled six pure ECs (metastatic and non-metastatic) with methylated DNA immunoprecipitation (MeDIP) followed by DNA-tiling hybridization (using Human Tiling Array 2.0R Chips), identifying hypermethylated DMRs in this tumor subtype, including X- and Y-linked genes and others related to metabolism. Follow-up on these studies might lead to identification of biomarkers for diagnosing this aggressive tumor subtype.

More recently, an integrated analysis of TGCTs was produced by combining high-dimensional ‘omics’ assays (genomics, epigenomics, transcriptomics, and proteomics) [127]. This study included 137 TGCTs from 133 patients and confirmed the remarkably distinct global methylation status (using the above mentioned 450K array) of different histological subtypes of TGCTs, again employing LC for methylation (after which the methylated peak was maintained in NSTs, while the intermediate methylation peaks disappeared in SEs). The study also depicted that ECs display methylation at the so-called CpH sites (non-canonical cytosine sites), a finding that even correlated with the amount of EC component, in line with the data of Killian et al. [88]. Both SEs (especially those with *KIT* mutations) and the majority of NSTs disclosed lack of methylation at imprinting sites. Finally, the analyses identified methylation of tumor suppressor genes already explored in other studies (see below), such as *BRCA1*, and recognized *RAD51C* silencing in NSTs. As both genes are implicated in homologous recombination (HR) DNA repair pathway, these findings seem to indicate a role of this pathway in TGCTs.

#### 3.2.2. MicroRNAs Relating to the Developmental Model

Control of pluripotency, early development and, subsequently, the development of GCNIS cells implies the expression of a number of biomarkers which may ultimately be used for diagnostic purposes, including messenger RNAs (mRNAs), protein players and also ncRNAs, such as miRs [128]. miRs are part of the small ncRNA (sncRNA) family, which means they are composed of less than 200 nt. Being the most studied, biologically relevant and versatile ncRNAs, they play important roles in many physiological and cancer-related processes, and they do so by dynamically regulating gene expression [129,130]. RNA-based biomarkers such as miRs display many advantages over protein-based ones, namely higher sensitivity and specificity (PCR techniques versus antibody-based techniques) and lower cost (as each protein requires a different antibody); and also over DNA-based biomarkers, as they are able to reflect dynamic cellular states and some of them are able to circulate stably in plasma and/or serum [131]. This has attracted attention towards miRs as biomarkers suitable for testing in liquid biopsies of cancer patients, for both diagnostic, prognostic, and predictive purposes [132]. This is particularly important in TGCTs, as commonly used serum markers (beta subunit of the human chorionic gonadotropin (β-HCG), alpha-fetoprotein (AFP), and lactate dehydrogenase (LDH)) have limited diagnostic performance, especially in certain tumor subtypes such as SE and EC [133]. In this vein, significant effort has been invested in translating miR testing to the clinics, hence the numerous publications in the field (Figure 3).

Voorhoeve et al. [134] were pioneers in identifying miRs that function as oncogenes (oncomiRs) in TGCTs by use of a miR library; they uncovered miR-372 and miR-373 role in neutralizing p53 function, by directly inhibiting the expression of the *LATS2* tumor suppressor. This mechanism constitutes an explanation for tumor progression in TGCTs, which typically show absence of p53 mutations (i.e., they harbor wild-type p53). Gillis et al. [135] pursued a high throughput screen of 156 miRs in GCT tissues and cell lines, confirming the relevance of the miR-371–373 cluster. Also, they demonstrated variations in miR expression relating to degree of maturation, again establishing a parallelism between GCTs development and embryogenesis, allowing for discrimination of tumor subtypes. Another study also profiled 615 miRs in GCTs, again confirming the value of miR-371~373 and miR-372 clusters, which were elevated in all histological subtypes, both in adult and pediatric patients [136]. These miRs were also documented to be overexpressed in GCNIS tissues [137]. Vilela-Salgueiro et al. [138] also examined TGCT tissues and showed miR-371a-3p discriminated TGCTs from normal testicular parenchyma with high sensitivity and specificity, rendering an area under the curve (AUC) of 0.99. Additionally, higher expression levels of this miR was apparent in SEs compared to NSTs, and also among individual NST subtypes, with tumor samples showing a decreasing expression of miR-371a-3p in parallel with tumor differentiation. The authors have also showed that miR-371a-3p was able to discriminate TE tissue samples (including pure TE and TE component isolated from mixed tumors, with diverse degrees of maturation) from normal testis, although in serum this could not be confirmed so far, which might impair its use as a biomarker, particularly in predicting residual pure TE after chemotherapy (see below) [138,139,140].

Expression of these promising miR clusters was early pursued in serum and plasma samples. Indeed, Murray et al. [141] soon documented that the most relevant members of the miR-371~373 and miR-372 clusters were elevated at diagnosis in the serum of a four-year-old boy with a YST, and that levels decreased after surgery and chemotherapy, opening the way for exploring these biomarkers in this line [142,143]. Indeed, since then, numerous publications exploring the potential of these miR clusters have been published and extended the findings of this sole case report into large cohorts of patients representative of various tumor entities [144,145]. Dieckmann et al. [146] further innovated by measuring miR-371-3 directly from testicular vein blood, which directly drains from the tumor; not surprisingly, the latter samples showed higher expression levels when compared to cubital vein-derived blood. The authors also measured the expression of this miR in tissues, but levels showed no correlation with the ones in serum. This was also reported by the Looijenga group [128,147]. The clinical utility of measuring serum miR-371a-3p was, hence, demonstrated [148].

Methodologies for detection of these miRs in serum evolved; Gillis et al. [147] used the targeted serum miR (TSmiR) test in GCTs and healthy control males, and showed overexpression of miR-371/372/373/367 in GCT patients, which decreased towards baseline after treatment, disclosing 98% sensitivity. Also, there was a trend for higher expression levels of these miRs in patients with metastases, pointing towards an association with tumor burden. Soon, novel miRs were also uncovered by high-throughput methods, including miR-511, -26b, -769, -23a, -106b, -365, -598, -340, and let-7a [149]. Spiekermann et al. [150] also described that miR-371-3 expression levels might be assessed without the need for an endogenous control if experiments were performed under controlled conditions. Other bodily fluids were soon tested for miR-371a-3p expression levels: high expression levels were depicted in pleural effusions and seminal plasma of patients, but not in urine samples [151]. However, another study showed that miR-142 was upregulated in seminal plasma from TGCT patients, but that miR-371-3 (along with miR-34b) were downregulated compared to controls, contrarily to the previous study [152]. High expression levels were also depicted in hydrocele fluid surrounding tumors [153] and also in cerebral spinal fluid (CSF) of pediatric patients with extragonadal malignant GCTs, allowing for discrimination from intracranial non-GCTs and allowing detection of relapses with high sensitivity and specificity [154]. To date, the miR with the best overall performance in serum samples was shown to be miR-371a-3p (with a sensitivity of 88.7% and specificity of 93.4%); also, expression levels correlated with relapse and dropped after treatment [155], and may additionally be used for detecting patients with GCNIS, possibly guiding the decision to perform a testicular biopsy in this context [156]. miR-371a-3p expression was also shown to outperform classical serum markers in detecting disease relapse, except in TEs [140,157].

The largest series evaluated thus far identified (with ampTSmiR test) miR-371a-3p, 373-3p, and 367-3p as highly sensitive and specific in TGCTs diagnosis (joint AUC of 0.96) [139]. More recently the work of Leão et al. [158] provided answers to a clinically relevant scenario, which is the detection of residual disease post-chemotherapy, as there are no validated markers for predicting viable disease and up to 50% of cases show only fibrosis and necrosis. This way, a biomarker for predicting viable disease is desirable, perhaps avoiding unnecessary surgery and related morbidity. In their work, it was shown that miR-371a-3p discriminated viable disease with an AUC of 0.87. Further studies on this subject are, however, necessary for clinical implementation to take place. Finally, Radtke et al. [159] analyzed stage I TGCT patients and demonstrated that miR-371a-3p has a very short half-life (less than 12 h), decaying very rapidly in the following three days after orchiectomy. Also, Mego et al. [160,161] reported the clinical utility of using miR-371a-3p plasma levels in predicting patient outcome in a population of TGCT patients set to be submitted to first line chemotherapy (higher levels associating with higher tumor burden and disease extent and negative levels culminating in better overall and progression-free survival).

All in all, miRs have shown great potential as biomarkers of both type I and type II GCTs, and also of GCNIS, and they are promising means for diagnosing and monitoring these patients. Their possible role as therapeutic targets is still largely unexplored and might constitute a promising way of avoiding cytotoxic treatments and its long-term side effects in such young patients [18]. A recent paper from Salvatori et al. [162] has used a mouse model for showing that plasma levels of miRs 371/302/C19MC were accurate in detecting undifferentiated and malignant components arising in xenografts derived from mice injected with human pluripotent stem cell (hPSC) lines and human malignant GCT cell lines. This study supports that these miRs might predict the emergence of malignancy in patients undergoing transplantation of hPSCs as a means of therapy, which thus far could only be determined by classical histological evaluation of TEs [163,164,165].

Besides methylation-related data, the already mentioned integrated study [127] depicted differentially expressed miRs (miR-sequencing (miR-seq) data) among tumor subtypes: it introduced the miR-519 cluster as being overexpressed in ECs (probably negatively regulating transcripts in this tumor subtype) and further confirmed miR-371a-3p value as a TGCT biomarker across subtypes (with the possible exception of TEs). It also disclosed miR-375 overexpression in TEs and YSTs, but not in ECs and SEs, meaning that it might complement miR-371a-3p expression assessment, especially in serum samples.

A summary of the mentioned studies on TGCT biomarkers regarding both methylation and miRs is depicted in Table 4.

## 4. Conclusions

An overview of TGCTs from a developmental perspective was provided. This model underlies the rationale for continuing to uncover epigenetic biomarkers that can be translated to the clinical practice. This selection is powerful because of the consistency of the identified factors in crucial regulatory pathways during embryogenesis, retained in TGCTs, and GCTs in general.

An integrated model (Figure 4) for defining TGCTs as distinct subtypes, concerning both genetic, cytogenetic, and epigenetic biomarkers, is warranted. We have showed that both DNA methylation profiles and miRs expression differ greatly among histological TGCT subtypes, and their detection in liquid biopsies has proved its use, such as miR-371a-3p. Somatic mutations are scarce in TGCTs, and are present mainly in SE components, especially those concerning *KIT* (which define a specific subset of SEs). Extensive aneuploidy (and frequent presence of i(12p)) is a hallmark of TGCTs, regardless the histologic type. Only by integrating all these factors can we reveal novel unappreciated diversity within TGCTs as clinical entities.

The use of novel high-throughput techniques will surely continue to open the door for characterizing new epigenetic aberrations in TGCTs, which can help us better understand their biology and, ultimately, be used for diagnostic and follow-up purposes.

## Figures and Tables

**Figure 1 ijms-20-00258-f001:**
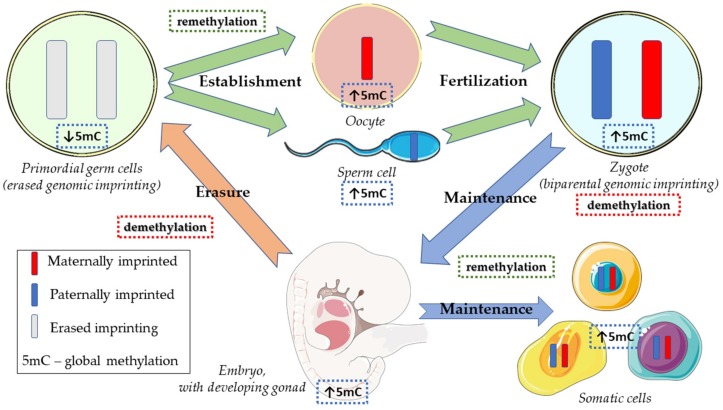
Cycle of genomic imprinting and global methylation.

**Figure 2 ijms-20-00258-f002:**
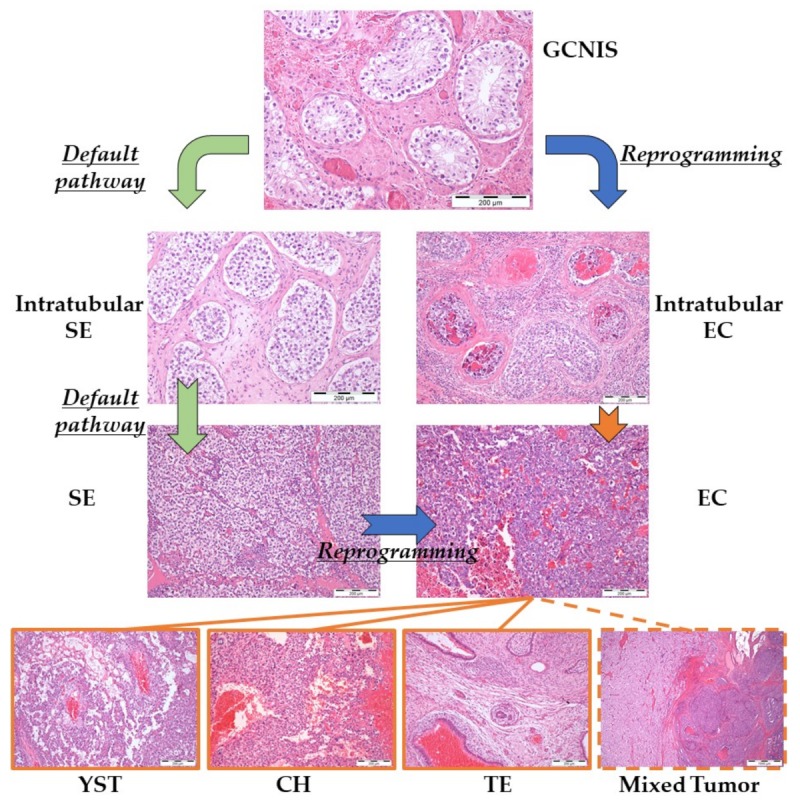
Pathogenesis of type II testicular germ cell tumors. Abbreviations: CH—choriocarcinoma; EC—embryonal carcinoma; GCNIS—germ cell neoplasia in situ; SE—seminoma; TE—teratoma; YST—yolk sac tumor.

**Figure 3 ijms-20-00258-f003:**
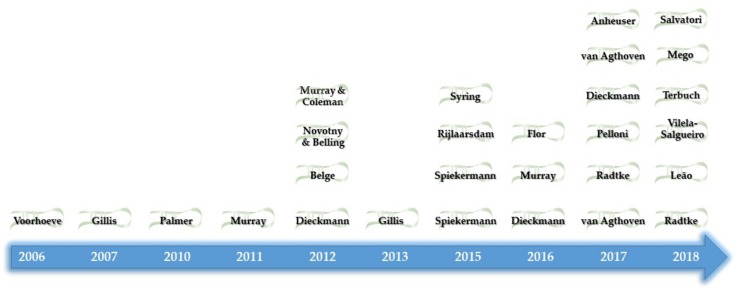
Chronological view of most relevant publications regarding microRNAs in testicular germ cell tumors (see text for details).

**Figure 4 ijms-20-00258-f004:**
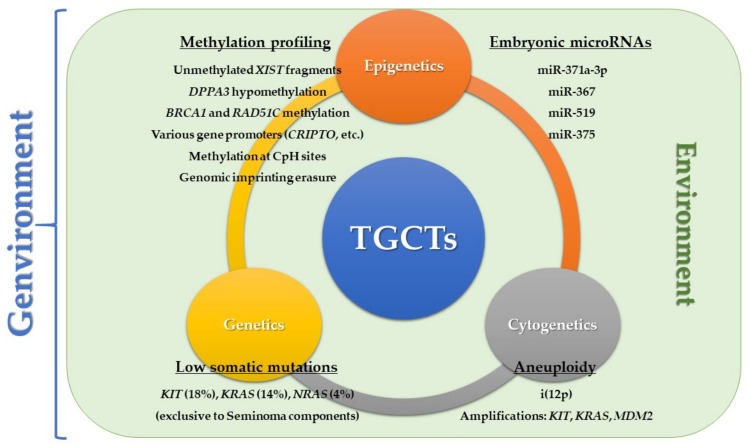
Integrative view of the genvironmental model, with focus on genetic, cytogenetic, and epigenetic factors, which are continuously modified and conditioned by the surrounding environment, ultimately determining the cell fate and tumor progression (see text for details).

**Table 1 ijms-20-00258-t001:** Epidemiology of germ cell tumors: incidence, prevalence, and mortality data.

Statistics	Context	Source
Age adjusted incidence rates: 64/1,000,000 (males) versus 4/1,000,000 (females)	Germ cell tumors	Europe (EUROCARE)
Incidence rates: 0.8% rise/year Estimated new cases: 5.7/100,000/year (all males, 2011–2015)	Testicular cancer	United States of America (SEER)
Age-standardized incidence rate: 1.7/100,000 (all males) versus 2.7/100,000 (males aged 15–39 years) 5-year prevalence: 150,377 cases (males aged 15–39 years) Estimated new cases (85,635) and deaths (13,288) in 2040 (all males)	Testicular cancer	World (Globocan)

**Table 2 ijms-20-00258-t002:** Genetic and environmental risk factors for germ cell tumors.

Factor	Relative Risk OR
Genetic	
Familial risk Brother with TGCT Father with TGCT	8–10 xs 4–6 xs
Studies in twins Monozygotic twins Dizygotic twins	76 xs 35 xs
Contralateral tumor	24.8–27.6
Various SNPs KITLG-related	OR >2 or <0.5
Environmental	
Internal	
Cryptorchidism	3.5–17.1
Infertility	1.16–6.72
Hypospadias	1.26–3.61
Atrophy	20.5
Previous inguinal hernia	1.63
Microlithiasis	3.42–13.2
Disturbed hormonal conditions in utero (maternal bleeding, first born child, low and high birthweight, short gestational age)	~1.3
Low birthweight	OR 1.28
Number of siblings ≥5	OR 0.71
External	
High body mass index	↑/↓/-
High stature	↑/-
Late onset of puberty	↓
Diet high in fat and dairy products	↑
Low physical exercise	↑/↓/-
Firefighters, metal/leather/agricultural workers	↑
Testicular trauma	↑
Marijuana smoking	OR 1.7

Abbreviations: KITLG—KIT-ligand; OR—odds ratio; TGCT—testicular germ cell tumor.

**Table 3 ijms-20-00258-t003:** Proposed classification of testicular germ cell tumors, according to their developmental state (adapted from [50]).

GCT Type	Age Group	Sex	Site	Phenotype	Developmental State	Precursor Cell	GI
0	Neonates	F/M	Midline	Included and parasitic twins	Omnipotent (2C state)	Blastomere	BiP
I	<6 years	F/M	Gonads, midline	TE, YST	Pluripotent (primed state)	Methylated PGC/gonocyte	BiP to partially erased
II	Postpubertal	>>M	Gonads, midline	SE/Dysg, NST	Totipotent (naïve state)	Hypomethylated PGC/gonocyte	Erased
III	>55 years	M	Testis	ST	Spermatogonium to premeiotic spermatocyte	Spermatogonium/spermatocyte	Partially to complete paternal
IV	Postpubertal	F	Ovary	Dermoid cyst	Maternally imprinted 2C state	Oogonia / oocyte	Partially to complete Maternal
V	Postpubertal	F	Placenta, uterus	Hydatidiform mole	Paternally imprinted 2C state	Empty ovum / spermatozoa	Completely paternal
VI	>60 years	F/M	Ovary, atypical sites	Resembling type I or NST components of type II	Primed state or NST lineages of naïve state	Somatic cell induced to pluripotency	Pattern of originating cell

Abbreviations: BiP—biparental; Dysg—dysgerminoma; F—female; GCT—germ cell tumor; GI—genomic imprinting; M—male; NST–non-seminomatous tumors; PGC—primordial germ cell; SE—seminoma; ST—spermatocytic tumor; TE—teratoma; YST—yolk sac tumor.

**Table 4 ijms-20-00258-t004:** Summary of studies on testicular germ cell tumor biomarkers regarding methylation and microRNAs.

Methodology	Sample Type	Major Findings	Year	Author
Methylation				
Bisulfite sequencing; PCR	Tissues (n = 31 TGCTs) and plasma (n = 25 TGCT samples, n = 24 non-TGCT samples)	XIST region IV frequently unmethylated in TGCTs, especially in SEs	2004	Kawakami et al.
Bisulfite sequencing; COBRA	Tissues (n = 14 TGCTs, n = 10 adjacent testicular parenchyma, n = 3 non-TGCTs) and TGCT cell lines	*LINE1* hypomethylated in both SEs and NSTs; *XIST* and *CDH1* mainly hypomethylated in SEs and methylated in NSTs	2011	Ushida et al.
qMS-PCR	Tissues (n = 161 TGCTs, n = 16 controls)	Differential methylation of *CRIPTO, HOXA9, MGMT, RASSF1A* and *SCGB3A1* gene promoters among TGCT subtypes	2018	Costa et al.
Genome-wide DNA methylation analysis	Tissues (n = 130 TGCTs, n = 128 benign neighboring testes)	*DPPA3* is hypomethylated in both SEs and NSTs; hypermethylation of *HM13* in NSTs and subtype-specific hypermethylation of *H19* in TEs	2016	Killian et al.
Genome-wide DNA methylation analysis	Tissues (n = 91 GCTs) and GCT cell lines	SEs, dysgerminomas and STs are globally hypomethylated, while ECs, NSTs and type I TEs are hypermethylated	2015	Rijlaarsdam et al.
Genome-wide DNA methylation analysis; RT-qPCR	GCT cell lines	Localized hypermethylation status in YSTs vs. disperse hypermethylation status in ECs and TEs	2015	Noor et al.
MeDIP; DNA-tiling hybridization; RT-qPCR; IHC	Tissues (n = 6 ECs)	Hypermethylated DMRs in ECs (X- and Y-linked genes, genes related to metabolism)	2016	Cheung et al.
Genome-wide DNA methylation analysis	Tissues (n = 137 TGCTs)	ECs display methylation at CpH sites; methylation of *BRCA1* and *RAD51C* silencing in NSTs	2018	Shen et al.
**MicroRNAs**				
miR library	NA	miR-372 and miR-373 netralize p53 (oncomiRs)	2006	Voorhoeve et al.
High-throughput screening of 156 miRs; qPCR	GCT tissues (n = 69) and cell lines	Relevance of miR-371–373 cluster; association with differentiation	2007	Gillis et al.
High-throughput screening of 615 miRs; RT-qPCR	Pediatric malignant GCTs, controls and GCT cell lines (n = 48)	Overexpression of miR-371~373 and miR-372 clusters in all tumor subtypes	2010	Palmer et al.
Multiplex PCR	Serum (n = 1) of a four-year-old boy	First report of utility of serum miRs in GCTs (miR-371–373 and miR-302 clusters); decrease after treatment	2011	Murray et al.
RT-qPCR	Serum (n = 12 patients, n = 11 controls)	Overexpression of miR-371-3 in patients and decrease after treatment	2012	Belge et al.
RT-qPCR	Serum (n = 8 malignant GCTs)	Additional specificity of using miR-367-3p	2012	Murray and Coleman
RT-qPCR	Serum (n = 24 GCTs, n = 17 controls) and GCT tissues (n = 15)	miR-371~373 measured in TVB in 6 patients (higher levels); no correlation with levels in tissues	2012	Dieckmann et al.
miR array; RT-qPCR	GCNIS tissue samples (n = 12)	Identification of miRs unique to GCNIS cells	2012	Novotny and Belling et al.
TSmiR	Serum (n = 80 GCTs, n = 47 controls, n = 12 non-GCT masses)	miR-371/372/373/367 panel with 98% sensitivity in diagnosis; higher expression levels in metastatic patients	2013	Gillis et al.
RT-qPCR	Serum (testing cohort: n = 30 patients and n = 18 controls; validation cohort: n = 76 patients, n = 84 controls)	miR-367-3p, miR-371a-3p, miR372-3p and miR-373-3p overexpressed in patients; miR-371-a-3p showing 84.7% sensitivity and 99% specificity in diagnosis	2015	Syring et al.
RT-qPCR	Serum (n = 25 GCTs, 6 GCNIS, n = 24 non-testicular malignancies, n = 20 controls), seminal plasma (n = 5), urine (n = 3) and pleural effusions (n = 1)	miR-371a-3p detected in seminal plasma and pleural effusions, but not in urine; confirmation of its value in serum	2015	Spiekermann et al.
High-throughput screening of 750 miRs; RT-qPCR	Serum (n = 14 GCTs, n = 11 controls)	Confirmation of the relevance of miR-371–373 cluster; novel relevant miRs identified	2015	Rijlaarsdam et al.
RT-qPCR	Serum (n=25 TGCTs, n=4 non-TGCTs, n = 17 controls)	Suggestion that normalization (relative quantification) is not required when quantifying miR-371-3	2015	Spiekermann
RT-qPCR	Serum and cerebral spinal fluid (n = 45 each) of 25 pediatric patients	Four serum microRNA panel (miR-371a-3p, miR-372-3p, miR-373-30 and miR-367-3p) with high sensitivity and specificity in discriminating intracranial GCT vs. non-GCT malignancies; first demonstration of relapse detection	2016	Murray et al.
RT-qPCR	GCT tissues and serum (n = 25 patients)	C19MC cluster overexpressed in aggressive subtypes	2016	Flor et al.
RT-qPCR	Tumor surrounding hydroceles (n = 9) and serum (n = 64 GCTs)	Hydroceles showing high levels of miR-371a-3p; association with tumor size; confirmed the value of miR-371a-3p in follow-up (relapse detection)	2016	Dieckmann et al.
ampTSmiR	Serum (n = 250 TGCTs, n = 60 non-TGCTs, n = 104 controls)	Largest series tested; panel composed of miR-371a-3p, miR-373-3p and miR-367-3p with 90% sensitivity and 91% specificity	2017	van Agthoven et al.
RT-qPCR	Serum (n = 312 consecutive patients with testicular disease)	Elevated levels aided in detection of clinically silent GCTs and metastases	2017	Anheuser et al.
RT-qPCR	Serum and seminal plasma (n = 48 patients, n = 28 controls)	miR-371a-3p suggested as a poor biomarker in seminal plasma, contrarily to miR-142	2017	Peloni et al.
RT-qPCR	Serum (n = 166 GCTs, n = 106 controls)	miR-371a-3p shows the best performance in TGCT detection (88.7% sensitivity, 93.4% specificity)	2017	Dieckmann et al.
RT-qPCR	Serum (n = 27 GCNIS)	miR-371a-3p overexpressed in GCNIS patients	2017	Radtke et al.
ampTSmiR	Serum (n = 1 SE, n = 5 NST) of patients with relapse/residual disease	miR-371a-3p outperformed classical protein markers in detection of disease relapse, except for mature TE	2017	van Agthoven et al.
RT-qPCR	Tissues (n = 119 TGCTs, n = 15 controls)	miR-371a-3p discriminated TGCTs from controls with 92% sensitivity and 93% specificity; decreasing expression with tumor differentiation; TEs discriminated from controls	2018	Vilela-Salgueiro et al.
ampTSmiR	Serum (n = 82 TGCTs)	miR-371a-3p discriminates viable disease post-chemotherapy (AUC = 0.87)	2018	Leão et al.
RT-qPCR	Serum (24 TGCTs, clinical stage I)	miR-371a-3p has a very short half-life (<12 h)	2018	Radtke et al.
RT-qPCR	Serum (n = 10 TGCT patients with relapse)	Confirmed miR-371a-3p value in detecting relapses	2018	Terbuch et al.
ampTSmiR	Plasma (n = 199 TGCTs, before chemotherapy)	miR-371a-3p predicts prognosis in chemotherapy naïve patients	2018	Mego et al.
Teratoma assay (mouse model)	Plasma of mice	Value of miR-371 family members in detecting undifferentiated and potentially malignant elements present in xenografts	2018	Salvatori et al.
miR-sequencing data	Tissues (n = 137 TGCTs)	miR-519 cluster overexpressed in ECs; miR-375 overexpressed in TEs and YSTs	2018	Shen et al.

Abbreviations: AR—androgen-receptor; AUC—area under the curve; COBRA—combined bisulfite restriction analysis; DMR—differentially methylated region; EC—embryonal carcinoma; GCTs—germ cell tumors; MeDIP—methylated DNA immunoprecipitation; miR—microRNA; NST—non-seminomatous tumor; qMS-PCR—quantitative methylation-specific polymerase chain reaction; RT-qPCR—real-time quantitative polymerase chain reaction; SE—seminoma; ST—spermatocytic tumor; TE—teratoma; TGCTs—testicular germ cell tumors; TVB—testicular vein blood; YST—yolk sac tumor.

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
