# Peer review of "Human Germ Cell Tumors are Developmental Cancers: Impact of Epigenetics on Pathobiology and Clinic"

_ijms, 2019, doi:10.3390/ijms20020258_

Reviewer 1 Report

The review article titled “The pathobiological and clinical impact of epigenetics on human germ cell tumors as a developmental cancer”, presents a unique perspective on the underlying cause of human germ cell tumors (GCTs), with emphasis on testicular GCT (TGCT). Specifically, authors argue that metastatic GCTs arise from an interplay between epigenetic (e.g., imprinting), developmental, and environmental factors, and not attributed to somatic abnormalities. They have meticulously elaborated on genetic factors, like, deregulated parental genetic imprinting, defects in germline programming, and developmental abnormalities in gonadal locale/maturation, as critical mediators in tumor causation. In addition, the review also details the possible utilization of two epigenetic factors – methylation patterns and miRNAs as possible biomarkers. However, there are a few issues with the manuscript as mentioned below.

Major Comments

1.      The ‘Genvironmental’ Model (section 1.2) should be clearly described under genetic and environmental factors separately first (under distinct sub-heading), and then as an interplay between the two. In addition, the environmental factors need to categorized clearly into external environment (postnatal factors like exposure to diet, physical exercise, marijuana etc.), vs. internal developmental/medical factors, like, cryptorchidism. This section is presented in a very confusing manner.

2.      Section (3.2) detailing the utilization of methylation patterns and miRNAs as possible biomarkers should additionally be represented in a tabular format, for convenience.

Minor Comments

1.      The title of the manuscript is confusing. It needs to be clear. Similarly, “a developmental cancer” in the title for section 2 is confusing.

2.      All the abbreviations used in the main manuscript should be described in a tabular format separately.

3.      Individual rows in Table 1 should be separated with a line in between.

4.      The abbreviation GCNIS first appears in Line# 117. However, its is described later in Line# 146. It should be described earlier.

5.      The epidemiology statistics (Section 1.1) should additionally be presented in a tabular format.

6.      The text in Figure 3 is not legible enough due to the miRNA watermark.

7.      Line# 227 – “In practice, this unifying model for GCTs…..”. Replace "means they can be" by "enables them to be".

8.      Line# 254 – replace the word “progress” with a suitable term or rephrase.

9.      Line# 539 – provide appropriate reference9s) for the histological evaluation of TEs.

Author Response

Reviewers' comments:

Reviewer #1:

The review article titled “The pathobiological and clinical impact of epigenetics on human germ cell tumors as a developmental cancer”, presents a unique perspective on the underlying cause of human germ cell tumors (GCTs), with emphasis on testicular GCT (TGCT). Specifically, authors argue that metastatic GCTs arise from an interplay between epigenetic (e.g., imprinting), developmental, and environmental factors, and not attributed to somatic abnormalities. They have meticulously elaborated on genetic factors, like, deregulated parental genetic imprinting, defects in germline programming, and developmental abnormalities in gonadal locale/maturation, as critical mediators in tumor causation. In addition, the review also details the possible utilization of two epigenetic factors – methylation patterns and miRNAs as possible biomarkers.

However, there are a few issues with the manuscript as mentioned below.

Major Comments

1.      The ‘Genvironmental’ Model (section 1.2) should be clearly described under genetic and environmental factors separately first (under distinct sub-heading), and then as an interplay between the two. In addition, the environmental factors need to be categorized clearly into external environment (postnatal factors like exposure to diet, physical exercise, marijuana etc.), vs. internal developmental/medical factors, like, cryptorchidism. This section is presented in a very confusing manner.

Reply:  Based on the well appreciated suggestion of this reviewer, we subdivided this section into three subsections, first dealing with genetic factors, second with environmental factors (including both external and internal), followed by an integrated section combining all. We strongly agree that this improved the readability of this part of the review.

2.      Section (3.2) detailing the utilization of methylation patterns and miRNAs as possible biomarkers should additionally be represented in a tabular format, for convenience.

Reply: We thank the Reviewer for his/her suggestion, which was also mentioned by Reviewer #2. We have added Table 4 summarizing these aspects discussed in the text (see below in the reply to Reviewer #2).

Minor Comments

1.      The title of the manuscript is confusing. It needs to be clear. Similarly, “a developmental cancer” in the title for section 2 is confusing.

Reply: The original title “The pathobiological and clinical impact of epigenetics on human germ cell tumors as a developmental cancerhas been adjusted to “Human germ cell tumors are developmental cancers: impact of epigenetics on pathobiology and clinic.”

Also, the title of section 2 was adjusted to “Pathobiology of Germ Cell Tumors and their developmental potential”.

2.      All the abbreviations used in the main manuscript should be described in a tabular format separately.

Reply: We thank the Reviewer for the suggestion. It has been added at the end of the Manuscript.

3.      Individual rows in Table 1 should be separated with a line in between.

Reply: We thank the Reviewer for his/her suggestion. It has been changed.

4.      The abbreviation GCNIS first appears in Line# 117. However, its is described later in Line# 146. It should be described earlier.

Reply: It has been corrected, accordingly.

5.      The epidemiology statistics (Section 1.1) should additionally be presented in a tabular format.

Reply: We thank the Reviewer for his/her suggestion, which will make the paper easier to read. We have added Table 1, accordingly. We decided to summarize on the Table only the figures concerning incidence, prevalence and mortality, so not to repeat every figure stated in the text right beneath it.

Table 1. Epidemiology of germ cell tumors: incidence, prevalence and mortality data

Statistics

Context

Source

Age   adjusted incidence rates: 64/1,000,000 (males) versus 4/1,000,000 (females)

Germ   cell tumors

Europe   (EUROCARE)

Incidence   rates: 0.8% rise/year

Estimated   new cases: 5.7/100,000/year (all males, 2011-2015)

Testicular   cancer

United   States of America (SEER)

Age-standardized   incidence rate: 1.7/100,000 (all males) versus 2.7/100,000 (males aged 15-39   years)

5-year   prevalence: 150,377 cases (males aged 15-39 years)

Estimated   new cases (85,635) and deaths (13,288) in 2040 (all males)

Testicular   cancer

World   (Globocan)

6.      The text in Figure 3 is not legible enough due to the miRNA watermark.

Reply: We thank the Reviewer for calling our attention to this matter. It has been corrected by turning the miR watermark more transparent.

7.      Line# 227 – “In practice, this unifying model for GCTs…..”. Replace "means they can be" by "enables them to be".

Reply: It has been corrected in accordance.

8.      Line# 254 – replace the word “progress” with a suitable term or rephrase.

Reply: We have rephrased accordingly: “will become invasive tumors”.

9.      Line# 539 – provide appropriate references for the histological evaluation of TEs.

Reply: We thank the Reviewer for this suggestion. Proper references were added.

Reviewer 2 Report

This is an interesting article dealing with the contribution of epigenetic modifications to human germ cell tumors. There are however several sections of the manuscript that could be improved. The conclusions are really short and merely speculative: the authors must provide a more integrated overview of the described points.

Major points:

1) Section 1.2 The genvironmental model: authors should better explain the contribution of genetic and environmental factors related to TGCTs. They mention diet, physical exercise or environmental exposure as potential risk factors, but should clarify what kind of diet, physical activity or exposure. Similarly, in line 117 it is unclear what "GCNIS" is for. The genetics of TGCTs is briefly summarized. I would encourage the authors to include a Table and/or figure showing all the genetic and environmental risk factors linked to TGCT and provide the relative/risk or odds ratio for each of them.

2) Authors should include a paragraph explaining epigenetic mechanisms in details.

3) Section 3.1 The use of high-throughput... The authors describe epigenome-wide studies in this section. It is therefore unclear why these studies are not described in section 3.2 The role of epigenetics.

4) One or two Tables are required to summarize both genome-wide and candidate-gene methylation investigations, including the used techniques and the main findings of each study.

5) Also for section 3.2.2 a Table summarizing the studies is required.

6) Conclusions: the manuscript lacks an integrated overview of the use ob genetic, cytogenetic and epigenetic biomarkers for TGCT subtyping. Authors must be able to integrate the various articles described in the study and provide a discussion (as well as a diagram) of the major genetic, cytogenetic and epigenetic marks that could be used to stratify human germ cell tumors into different subtypes and clinical entities.

Author Response

Reviewer #2:

This is an interesting article dealing with the contribution of epigenetic modifications to human germ cell tumors.

Reply: We thank the Reviewer for his/her positive opinion on our work.

There are however several sections of the manuscript that could be improved. The conclusions are really short and merely speculative: the authors must provide a more integrated overview of the described points.

Major points:

1) Section 1.2 The genvironmental model: authors should better explain the contribution of genetic and environmental factors related to TGCTs. They mention diet, physical exercise or environmental exposure as potential risk factors, but should clarify what kind of diet, physical activity or exposure. Similarly, in line 117 it is unclear what "GCNIS" is for. The genetics of TGCTs is briefly summarized. I would encourage the authors to include a Table and/or figure showing all the genetic and environmental risk factors linked to TGCT and provide the relative/risk or odds ratio for each of them.

Reply: We thank the Reviewer for his/her suggestion, which was also pointed out by Reviewer #1. In order to comply with both Reviewers we have divided the section in the manuscript in subsections, described the influences of environmental factors in more detail and have added a Table 2 to summarize this section.

Table 2. Genetic and environmental risk factors for germ cell tumors

Factor

Relative Risk / OR

Genetic

Familial risk

    Brother with TGCT

    Father with TGCT

8-10xs

4-6xs

Studies in twins

    Monozygotic twins

    Dizygotic twins

76xs

35xs

Contralateral tumor

24.8-27.6

Various SNPs

    KITLG-related

OR >2 or<0.5< span="">

Environmental

Internal

    Cryptorchidism

3.5-17.1

    Infertility

1.16-6.72

    Hypospadias

1.26-3.61

    Atrophy

20.5

    Previous inguinal hernia

1.63

    Microlithiasis

3.42-13.2

    Disturbed hormonal conditions in utero (maternal bleeding, first   born child, low and high birthweight, short gestational age)

~1.3

    Low birthweight

OR 1.28

    Number of siblings >=5

OR 0.71

External

    High body mass index

//-

    High stature

/-

    Late onset of puberty

    Diet high in fat   and dairy products

    Low physical   exercise

//-

    Firefighters,   metal/leather/agricultural workers

    Testicular trauma

    Marijuana smoking

OR 1.7

Abbreviations: KITLG – KIT-ligand; OR – odds ratio; TGCT – testicular germ cell tumor

2) Authors should include a paragraph explaining epigenetic mechanisms in detail.

Reply: We thank the Reviewer for pointing this out. We have added such a paragraph:    Epigenetics encompasses an array of processes that change gene expression without altering the DNA sequence, leading to a change in phenotype without changing the genotype. It comprises covalent modifications of DNA (such as DNA methylation), histone variants, histone post-translational modifications and non-coding RNAs (ncRNAs). DNA methylation, one of the most studied mechanisms, occurs by addition of methyl groups to the fifth carbon of cytosines, occurring preferentially at CpG sites, which are unevenly distributed in the genome – being concentrated in the so-called CpG islands. Differential methylation of gene promoters ultimately affect gene expression. Similarly, a number of ncRNAs are involved in the dynamic and environmentally sensitive regulation of gene expression. These molecules are known to interact (directly or indirectly) with the other established epigenetic mechanisms and can also directly interfere with messenger RNA (mRNA); this way, they can be seen as an extension of the complex epigenetic network, establishing important bridges between related modifications and truly influencing gene expression [85-87]. In this review, we will be focusing on methylation and a subtype of ncRNAs – the microRNAs (miRs).

3) Section 3.1 The use of high-throughput... The authors describe epigenome-wide studies in this section. It is therefore unclear why these studies are not described in section 3.2 The role of epigenetics.

Reply: We thank the Reviewer for his/her suggestion. Accordingly, the studies on the section “Use of high-throughput methodologies” were discussed in “The role of epigenetics” section.

4) One or two Tables are required to summarize both genome-wide and candidate-gene methylation investigations, including the used techniques and the main findings of each study.

AND

5) Also for section 3.2.2 a Table summarizing the studies is required.

Reply: We thank the reviewer for his/her suggestion, which was also pointed out by Reviewer #1. In accordance, we have added a Table (Table 4) summarizing these studies. Only one Table, joining both methylation and miRNAs was added as the article contains already many “Figure and Table” elements.

Table 4. Summary of studies on testicular germ cell tumor biomarkers regarding methylation and microRNAs

Methodology

Sample type

Major findings

Year

Author

Methylation

Bisulfite   sequencing; PCR

Tissues   (n=31 TGCTs) and plasma (n=25 TGCT samples, n=24 non-TGCT samples)

XIST   region IV frequently unmethylated in TGCTs, especially in SEs

2004

Kawakami   et al

Bisulfite   sequencing; COBRA

Tissues   (n=14 TGCTs, n=10 adjacent testicular parenchyma, n=3 non-TGCTs) and TGCT   cell lines

LINE1   hypomethylated in both SEs and NSTs; XIST   and CDH1 mainly hypomethylated in   SEs and methylated in NSTs

2011

Ushida   et al

qMS-PCR

Tissues (n=161 TGCTs, n=16 controls)

Differential   methylation of CRIPTO, HOXA9, MGMT,   RASSF1A and SCGB3A1 gene   promoters among TGCT subtypes

2018

Costa   et al

Genome-wide   DNA methylation analysis

Tissues   (n=130 TGCTs, n=128 benign neighboring testes)

DPPA3 is   hypomethylated in both SEs and NSTs; hypermethylation of HM13 in NSTs and subtype-specific hypermethylation of H19 in TEs

2016

Killian   et al

Genome-wide   DNA methylation analysis

Tissues   (n=91 GCTs) and GCT cell lines

SEs,   dysgerminomas and STs are globally hypomethylated, while ECs, NSTs and type I   TEs are hypermethylated

2015

Rijlaarsdam   et al

Genome-wide   DNA methylation analysis; RT-qPCR

GCT   cell lines

Localized   hypermethylation status in YSTs vs disperse   hypermethylation status in ECs and TEs

2015

Noor   et al

MeDIP;   DNA-tiling hybridization; RT-qPCR; IHC

Tissues   (n=6 ECs)

Hypermethylated   DMRs in ECs (X- and Y-linked genes, genes related to metabolism)

2016

Cheung   et al

Genome-wide   DNA methylation analysis

Tissues   (n=137 TGCTs)

ECs   display methylation at CpH sites; methylation of BRCA1 and RAD51C   silencing in NSTs

2018

Shen   et al

MicroRNAs

miR   library

NA

miR-372   and miR-373 netralize p53 (oncomiRs)

2006

Voorhoeve   et al

High-throughput   screening of 156 miRs; qPCR

GCT   tissues (n=69) and cell lines

Relevance   of miR-371-373 cluster; association with differentiation

2007

Gillis   et al

High-throughput   screening of 615 miRs; RT-qPCR

Pediatric   malignant GCTs, controls and GCT cell lines (n=48)

Overexpression   of miR-371~373 and miR-372 clusters in all tumor subtypes

2010

Palmer   et al

Multiplex   PCR

Serum   (n=1) of a four-year-old boy

First   report of utility of serum miRs in GCTs (miR-371–373 and miR-302 clusters);   decrease after treatment

2011

Murray   et al

RT-qPCR

Serum (n=12 patients, n=11 controls)

Overexpression   of miR-371-3 in patients and decrease after treatment

2012

Belge   et al

RT-qPCR

Serum   (n=8 malignant GCTs)

Additional   specificity of using miR-367-3p

2012

Murray   and Coleman

RT-qPCR

Serum   (n=24 GCTs, n=17 controls) and GCT tissues (n=15)

miR-371~373   measured in TVB in 6 patients (higher levels); no correlation with levels in   tissues

2012

Dieckmann   et al

miR array; RT-qPCR

GCNIS   tissue samples (n=12)

Identification   of miRs unique to GCNIS cells

2012

Novotny   and Belling et al

TSmiR

Serum (n=80 GCTs, n=47 controls, n=12   non-GCT masses)

miR-371/372/373/367   panel with 98% sensitivity in diagnosis; higher expression levels in   metastatic patients

2013

Gillis   et al

RT-qPCR

Serum   (testing cohort: n=30 patients and n=18 controls; validation cohort: n=76   patients, n=84 controls)

miR-367-3p,   miR-371a-3p, miR372-3p and miR-373-3p overexpressed in patients; miR-371-a-3p   showing 84.7% sensitivity and 99% specificity in diagnosis

2015

Syring   et al

RT-qPCR

Serum   (n=25 GCTs, 6 GCNIS, n=24 non-testicular malignancies, n=20 controls),   seminal plasma (n=5), urine (n=3) and pleural effusions (n=1)

miR-371a-3p   detected in seminal plasma and pleural effusions, but not in urine;   confirmation of its value in serum

2015

Spiekermann   et al

High-throughput   screening of 750 miRs; RT-qPCR

Serum (n=14 GCTs, n=11 controls)

Confirmation of the relevance of miR-371–373   cluster; novel relevant miRs identified

2015

Rijlaarsdam   et al

RT-qPCR

Serum (n=25 TGCTs, n=4 non-TGCTs, n=17   controls)

Suggestion that normalization (relative   quantification) is not required when quantifying miR-371-3

2015

Spiekermann

RT-qPCR

Serum   and cerebral spinal fluid (n=45 each) of 25 pediatric patients

Four   serum microRNA panel (miR-371a-3p, miR-372-3p, miR-373-30 and miR-367-3p)   with high sensitivity and specificity in discriminating intracranial GCT vs non-GCT malignancies; first   demonstration of relapse detection

2016

Murray   et al

RT-qPCR

GCT   tissues and serum (n=25 patients)

C19MC   cluster overexpressed in aggressive subtypes

2016

Flor   et al

RT-qPCR

Tumor   surrounding hydroceles (n=9) and serum (n=64 GCTs)

Hydroceles   showing high levels of miR-371a-3p; association with tumor size; confirmed   the value of miR-371a-3p in follow-up (relapse detection)

2016

Dieckmann   et al

ampTSmiR

Serum   (n=250 TGCTs, n=60 non-TGCTs, n=104 controls)

Largest series tested; panel composed of   miR-371a-3p, miR-373-3p and miR-367-3p with 90% sensitivity and 91% specificity

2017

van   Agthoven et al

RT-qPCR

Serum   (n=312 consecutive patients with testicular disease)

Elevated levels aided in detection of clinically   silent GCTs and metastases

2017

Anheuser   et al

RT-qPCR

Serum and seminal plasma (n=48   patients, n=28 controls)

miR-371a-3p suggested as a poor biomarker in   seminal plasma, contrarily to miR-142

2017

Peloni   et al

RT-qPCR

Serum (n=166 GCTs, n=106 controls)

miR-371a-3p shows the best performance in TGCT   detection (88.7% sensitivity, 93.4% specificity)

2017

Dieckmann   et al

RT-qPCR

Serum   (n=27 GCNIS)

miR-371a-3p overexpressed in GCNIS patients

2017

Radtke   et al

ampTSmiR

Serum   (n=1 SE, n=5 NST) of patients with relapse/residual disease

miR-371a-3p outperformed classical protein markers   in detection of disease relapse, except for mature TE

2017

van   Agthoven et al

RT-qPCR

Tissues   (n=119 TGCTs, n=15 controls)

miR-371a-3p discriminated TGCTs from controls   with 92% sensitivity and 93% specificity; decreasing expression with tumor   differentiation; TEs discriminated from controls

2018

Vilela-Salgueiro   et al

ampTSmiR

Serum   (n=82 TGCTs)

miR-371a-3p discriminates viable disease   post-chemotherapy (AUC=0.87)

2018

Leão   et al

RT-qPCR

Serum   (24 TGCTs, clinical stage I)

miR-371a-3p has a very short half-life (<12h)< span="">

2018

Radtke   et al

RT-qPCR

Serum   (n=10 TGCT patients with relapse)

Confirmed miR-371a-3p value in detecting relapses  

2018

Terbuch   et al

ampTSmiR

Plasma   (n=199 TGCTs, before chemotherapy)

miR-371a-3p predicts prognosis in chemotherapy   naïve patients

2018

Mego   et al

Teratoma assay (mouse model)

Plasma   of mice

Value of miR-371 family members in detecting   undifferentiated and potentially malignant elements present in xenografts

2018

Salvatori   et al

miR-sequencing   data

Tissues   (n=137 TGCTs)

miR-519   cluster overexpressed in ECs; miR-375 overexpressed in TEs and YSTs

2018

Shen   et al

Abbreviations: AR – androgen receptor; AUC – area under the curve; COBRA – combined bisulfite restriction analysis; DMR – differentially methylated region; EC – embryonal carcinoma; GCTs – germ cell tumors; MeDIP - methylated DNA immunoprecipitation; miR – microRNA; NST – non-seminomatous tumor; qMS-PCR – quantitative methylation-specific polymerase chain reaction; RT-qPCR – real-time quantitative polymerase chain reaction; SE – seminoma; ST – spermatocytic tumor; TE – teratoma; TGCTs – testicular germ cell tumors; TVB – testicular vein blood; YST – yolk sac tumor

6) Conclusions: the manuscript lacks an integrated overview of the use of genetic, cytogenetic and epigenetic biomarkers for TGCT subtyping. Authors must be able to integrate the various articles described in the study and provide a discussion (as well as a diagram) of the major genetic, cytogenetic and epigenetic marks that could be used to stratify human germ cell tumors into different subtypes and clinical entities.

Reply: We thank the Reviewer for his/her suggestion. We have added such a paragraph and an integrative diagram:

“An integrated model (Figure 4) for defining TGCTs as distinct subtypes, concerning both genetic, cytogenetic and epigenetic biomarkers, is warranted. We have showed that both DNA methylation profiles and miRs expression differ greatly among histological TGCT subtypes, and their detection in liquid biopsies has proved its use, such as miR-371a-3p. Somatic mutations are scarce in TGCTs, and are present mainly in SE components, especially those concerning KIT (which define a specific subset of SEs). Extensive aneuploidy (and frequent presence of i(12p)) is a hallmark of TGCTs, regardless the histologic type. Only by integrating all these factors can we reveal novel unappreciated diversity within TGCTs as clinical entities”. 

Round  2

Reviewer 2 Report

The authors have nicely addressed my criticisms